# Fatty Acid Profile and Antioxidant Capacity of Dabai (*Canarium odontophyllum* L.): Effect of Origin and Fruit Component

**DOI:** 10.3390/molecules27123840

**Published:** 2022-06-15

**Authors:** Shanti Faridah Salleh, Olaide Olawunmi Ajibola, Crilio Nolasco-Hipolito, Ahmad Husaini, Carvajal Zarrabal-Octavio, Samuel Lihan, Gbadebo Clement Adeyinka, Firdaus R. Rosli, Idris Adewale Ahmed, Mohamed Zaky Zayed, Rosmawati Saat

**Affiliations:** 1Institute of Sustainable and Renewable Energy (ISuRE), University Malaysia Sarawak, Kota Samarahan 94300, Malaysia; sshanti@unimas.my; 2Faculty of Resource Science and Technology, University Malaysia Sarawak, Kota Samarahan 94300, Malaysia; haahmad@unimas.my (A.H.); firdaus.rizwan@gmail.com (F.R.R.); srosma@unimas.my (R.S.); 3Institute of Biodiversity and Environmental Conservation, University Malaysia Sarawak, Kota Samarahan 94300, Malaysia; lsamuel@unimas.my; 4Department of Biochemistry, Memorial University of Newfoundland, St. John’s, NL A1C 5S7, Canada; 5Institute of Biotechnology, University del Papaloapan, Circuito Central 200, Col. Parque Industrial, San Juan Bautista Tuxtepec 68301, Mexico; cnolasco@unpa.edu.mx; 6Biochemistry and Nutrition Chemistry Area, University of Veracruz, Juan Pablo II s/n, Boca del Rio 94294, Mexico; ocarvajal@uv.mx; 7Department of Chemical Engineering, Mangosuthu University of Technology, Durban 4031, South Africa; adeyinkagbadebo78.ga@gmail.com; 8Center for Natural Products Research and Drug Discovery, University Malaysia, Kuala Lumpur 50603, Malaysia; idrisahmed@um.edu.my; 9Department of Biotechnology, Faculty of Applied Science, Lincoln University College, Kelana Jaya, Petaling Jaya 47301, Malaysia; 10Forestry and Wood Technology Department, Faculty of Agriculture (EL-Shatby), Alexandria University, Alexandria 21527, Egypt; mzmohamedzaky86@gmail.com

**Keywords:** *Canarium odontophyllum*, cytotoxicity, antioxidant, fatty acid profiles, geographical location

## Abstract

In the present work, the influence of geographical location on the fatty acid profiles, antioxidant potential, as well as cytotoxicity of edible dabai fruit fractions (kernel, skin, and pulp) were analyzed. The fatty acid profiles were determined by Gas Chromatography (GC), and the antioxidant activity was quantified with free 2,2-diphenyl-1-picr/ylhdrazyl, while the cytotoxicity was assessed by the brine shrimp lethality test. The results showed that the samples from Sibu, Serian, and Kapit geographical locations had a high content of the saturated fatty acids, ranging from 46.63% to 53.31% in the three fractions. The highest mono-saturated fatty acids (MUFA) content was found in Sibu. Serian and Kapit kernel fractions MUFA, however, ranged from 21.2% to 45.91%. No fatty acid composition was detected in Bentong and Kanowit. The fatty acid composition and DPPH free radical scavenging antioxidant activity of dabai were statistically independent using a multivariate analysis in different localities in Malaysia. The skin fraction had a more appreciable antioxidant potential and toxicity level than the pulp and kernel fractions. The highest antioxidant activity (EC_50_ 198.76 ± 1.06 µg/mL) with an LC_50_ value of 1387.22 µg/mL was obtained from the Sibu skin fraction. Therefore, the fatty acid composition, antioxidant, as well as cytotoxicity analyses of the extracts from different localities indicated that “geographical location” remarkably influenced fatty acid composition, antioxidant activity, and toxicity.

## 1. Introduction

*Canarium odontophyllum* L. (Sibu olive) also known as dabai in Sarawak, Malaysia, is an evergreen tree from the Burseraceae family. It is one of the sources of local commercial fruits and local timber trees. The fruit is known as Kembayau in Brunei and Sabah [1]. It is a seasonal fruit with two peak production seasons: mainly July–August and November–December, depending on the weather pattern. In Southeast Asia Borneo Island, it is propagated from seeds [2].

Recently, Sibu olive has been reported to have the potential to be tapped for commercial purposes due to its medicinal and nutritional values. Hence, it is one of the most important sources of employment opportunities for most of the unemployed youths and women in the rural population [3]. During the past two decades, its cultivation has attracted considerable attention because of its general acceptability by consumers and its export potential [4]. The kernel and pulp of this fruit are rich in fat. The fat of the dabai kernel is highly saturated, while the fat of the dabai pulp is moderately saturated [5]. The highly saturated fat of the dabai kernel is the substitute for palm kernel fat and cocoa butter in making chocolates [6]. The fat derived from the dabai kernel and pulp has been biologically proven to enhance the lipid profile of laboratory rats treated with dabai fat [7]. The defatted dabai peel extract has also been reported to demonstrate protective properties, like lipid peroxidation and inhibition of oxidative stress [8].

The skin, flesh, and kernel of dabai have been reported to possess antioxidant properties [9]. According to Prasad et al. [10], the ethyl acetate fraction of dabai peel had stronger antioxidant capacities than other fractions. Besides antioxidants, the study also reported that the peel fraction had the highest phenolic and flavonoid contents compared to other fractions [10]. Stem bark and leaves of dabai have been shown to have weaker cytotoxicity against the human colorectal carcinoma (HCT) 116-cancer cell lines [11]. However, the previous study has shown that differences in climate, cultivar, and maturity exert an impact on the nutritional and physical properties of dabai fruits; a study had shown that dabai fruits have been proven to be an excellent source of lipids, protein, and energy, as well as certain essential minerals, such as calcium, magnesium, and phosphorus [12]. In addition, this fruit has high antioxidant properties and a creamy taste. Anthocyanin is the primary phenolic compound in dabai fruit. For example, the antioxidant abilities of different fruit parts (peel, pulp, and seeds) depend on the extraction method and the solvent used.

To the best of our knowledge, no published study is available on the influence of geographical location on the fatty composition, antioxidant activity, and toxicity of different edible fractions of dabai fruits. Furthermore, the origin effect has not been thoroughly studied, despite the reported effects of genotypes. Thus, this study aims to assess the effect of geographical location on the different edible fractions of dabai fruit (kernel, skin, and pulp) and to identify the best locality with the highest fatty composition, antioxidant properties, and non-toxicity potential among the different localities in Sarawak, Malaysia.

## 2. Materials and Methods

Matured dabai fruits from five localities in Sarawak, Malaysia, namely Sibu, Serian, Kapit, Kanowit, and Betong, were collected from the farm within these geographical locations. Fruits were cleaned under running tap water without any physical harm to remove any impurities or dirt. Fruits were allowed to dry overnight in the shade to avoid degradation of phytochemicals, before being stored in a freezer at a temperature of −20 °C. Frozen fruits were removed from the freezer and thawed at room temperature. Fruit parts, such as skin, pulp, and kernel, were separated manually. A knife was used to peel the skin fraction; the fraction of the pulp was scraped using a spoon, while the kernel fraction was collected using a sledgehammer to break the hard-shelled seeds. Prior to the analysis, the skin and pulp fractions were stored in the freezer at a temperature of about −20 °C, whereas the kernel fraction was dried in an oven for 6 h at a temperature of 30 °C before it was blended to form a powder [13]. 

### 2.1. Sample Extraction

A 100 g sample of each fraction of *C. odontophyllum* L. fruits (skin, pulp, and kernel) was extracted with n-hexane in a 2 L Erlenmeyer flask. The mixture was homogenized and soaked at an ambient temperature for 24 h before it was filtered through filter paper (TISH Scientific, Cleves, OH, USA). A rotary evaporator (Heidolph Laborota 4000, Heidolph Instruments GmbH & Co KG, Schwabach, Germany) was used to concentrate the extracted oil samples at 40 °C [14]. The extracted oil was flushed with nitrogen gas to remove the solvent’s residual before it was stored in a dark room. The extraction was done in triplicate.

### 2.2. Evaluation of the Fatty Acids Composition in the Extract Samples

The percentage of fatty acids from extracted oils in the skin, pulp, and kernel fractions of dabai from different geographical locations was qualitatively evaluated in this study. Briefly, the fatty acid standard was run using Gas Chromatography Mass Spectrometry (GC-MS) to know the retention time for individual fatty acid components (based on chromatogram peak), and this served as a reference for possible identification. The sample extract was then injected into GC-MS following the same procedure as done for the standard, a sample chromatogram was obtained, and each peak was identified using standard retention time. Each peak in the samples was further confirmed with the NIST mass spectral library, where 95 to 98% marge was considered. The percentage yield of individual fatty acid was thereby obtained.

A GC-MC Shimadzu brand (Model QP-2010 Plus) with an HP-5 fused capillary column (5% phenyl methylpolysiloxane stationary phase), with 30.0 m × 0.25 mm × 0.25 µm capacity, was used. The GC-MS configuration was programmed with the injector and detector temperature set at 200 °C and 210 °C, respectively. A split mode injection (1:10) was conducted. and helium was employed as a carrier gas at a controlled constant flow rate of 1.0 mL/min. The initial column oven temperature was 40 °C for 4 min, raised to 210 °C at the rate of 4 °C/min, and kept for a further 15 min. A 1.0 µL injection volume of each oil sample (dissolved with 200-µL n-hexane) was injected into the GC column. 

### 2.3. DPPH Free Radical Scavenging Antioxidant Activity

The method by Ajibola et al. [15,16] was adapted to determine 1, 1-diphenyl-2-picryl-hydrazyl (DPPH) free radical scavenging activities (RSA) of skin, pulp, and kernel fractions of dabai extracts. The extracts were first kept at 50 °C for 2 h, employing an orbital shaker incubator (New Brunswick C2, Edison, NJ, USA) with an agitation rate of 180 rpm. The homogenates were then filtered through filter papers employing Buchner funnels, before being freeze-dried and kept at −30 °C until further tests were determined. All the freeze-dried extracts were completely dissolved in 10.0 mL of methanol (CH_3_OH) and sonicated for 6 min to form a stock solution. The working standard solutions (10, 50, 100, 500, 1000 µg/mL) were further prepared from the stock solution. For each concentration, 1.0 mL of the solution was pipetted into a screw-top dark vial before it was thoroughly mixed with 3.0 mL of 0.1 mM solution of DPPH in CH_3_OH solution. The mixture was incubated in the dark at 25 °C for 30 min and analyzed at 517 nm employing a UV/vis spectrophotometer (Shimadzu Co., Kyoto, Japan), with the absorbance of 4 mL of 0.1 mM of DPPH in CH_3_OH solution, without the sample extract counted as a negative control and ascorbic acid as a positive control. Each solution was read in triplicate. The DPPH free radical scavenging activity (RSA) (%) was calculated according to Equation (1):(1)$$\mathrm{DPPH}\;\mathrm{free}\;\mathrm{radical}\;\mathrm{scavenging}\;\mathrm{activity}\;(\%)=\left(\frac{Ab_{0}-Ab_{1}}{Ab_{0}}\right)\times 100$$
where *Ab*_0_ = the absorbance of the control (without sample) at 517 nm;*Ab*_1_ = the absorbance of the sample at 517 nm


The RSA of the extracts was expressed in terms of EC_50_ value, which is the effective concentration by which 50% of DPPH free radical was scavenged. The EC_50_ value was determined from the sigmoid curve of antioxidant activity (%) versus the sample concentration; the lower the value of EC_50_, the stronger the extract has to act as a DPPH scavenger [17]. In comparison to both the extracts and ascorbic acid (positive standard), a standard antioxidant was prepared to find the extract with the strongest scavenging activity.

### 2.4. Brine Shrimp Lethality Bioassay

*Artemia salina* (brine shrimp) larvae were used to perform toxicity tests. About 0.7 g of *A. salina* egg was added into a conical beaker containing 1.0 L of seawater, with pH 7.6, temperature 26–28 °C, salinity 22 ppt, and continuous airflow for the hatching process to occur for 48 h [18]. For the brine shrimp lethality test, exactly 5.0 mg of lipid extracts was weighed in the calibrated sample bottle before it was dissolved in 5.0 mL of methanol. Exactly 10 ppm, 50 ppm, 100 ppm, 250 ppm, and 500 ppm of the solution were transferred into the test plates in triplicate. The solvent was evaporated to dryness in the fume hood before 0.2 mL of dimethyl sulfoxide (DMSO) and 4.8 mL of seawater were added to each of the six-well plates. Exactly 0.2 mL of the diluted sample was transferred into every well plate, and 10 nauplii (larvae) of *A. salina* cyst were added and incubated for 24 h under direct light at 25 °C. Three replicates per concentration were conducted in this biological activity. The solution containing 5.0 mg of thymol in 5.0 mL of seawater with 10 nauplii without the sample was used as a positive control (standard), whereas the solution containing 4.50 mL of seawater and 0.5 mL of DMSO with 10 nauplii without the sample was used as a negative control. During the study, no feed or air was needed since the feeding of brine shrimps with dry yeast suspension during the toxicity assessment was considered insignificant [19]. Populations of dead and surviving nauplii were counted in each well, and LC_50_ was calculated. Based on Thangapandi Veni calculations [19], toxicity characteristics of the extracts were estimated as follows (Equation (2)) [20]:(2)$$Mortality\;\left(\%\right)=\frac{F_{test}-D_{control}}{B_{control}}\times 100$$
where *F_test_* was the population of dead larvae in each test plate, *B_control_* was the population of live larvae in control plates, and *D_control_* was the population of dead larvae in each control plate.

### 2.5. Statistical Analysis

All results were expressed as group mean ± standard deviation (SD), analyzed using SPSS v. 21.0 (IBM statistical software 21.0). The statistical test was investigated using Tukey’s post hoc test, and the *p*-value < 0.05 was considered to be a significant difference. The determination of the mean death of the *A. salina* nauplii at different concentrations of the samples and the concentration that kills 50% (LC_50_) of the brine shrimp were made using Microsoft Excel, and the probit analysis test was conducted on statistical software SPSS (*p* < 0.05).

### 2.6. Multivariate Analysis

A Principal Component Analysis (PCA) based on a correlation matrix was performed to elucidate the correlated fatty acid composition and DPPH free radical scavenging antioxidant activity in different localities of dabai fruit. The results were further analyzed by a multivariate statistical approach employing a hierarchical cluster analysis (HCA, Pvclust function in R) associated with a proximity score matrix represented as a proximity heat-map.

## 3. Results and Discussion

### 3.1. Determination of Fatty Acids

The kernel and pulp of *C. odontophyllum* fruits, locally called Sibu olive and dabai, are rich in fat [5]. The highly saturated fat of the dabai kernel is not only used as a substitute for palm kernel fat and cocoa butter in making chocolates [6], but it has also been biologically employed to enhance the lipid profile of laboratory animals [7]. The antioxidant properties of the defatted dabai peel extract have also been reported [8]. Despite the potential of the dabai fruit, the origin effect has not been thoroughly studied, despite the reported effects of genotypes.

Figure 1 presents the GC-MS chromatogram for the mixture of FAME standards, whereby a total of 14 individual FAMEs were eluted. Table 1 presents the retention times for the identified components of the FAMEs standards, analyzed by GC-MS. 

Table 2, Table 3 and Table 4 present the total amount of fatty acids in the skin, pulp, and kernel fractions of dabai fruit obtained from different geographical locations, respectively. The dominant fatty acid was palmitic acid, ranging from 26.40% and 39.66%; followed by linoleic acid, ranging from 7% to 47.04%; oleic acid, ranging from 7% to 45.91%; and stearic acid, ranging from 9.01% to 23.42% (Table 2). Heptanoic acid was present in Kapit (skin) and Serian (pulp) in trace quantities, and eicosadienoic was present in Serian (skin) and Sibu (pulp). Furthermore, only the Serian sample contained a trace quantity of elaidic acid, whereas the Sibu sample was the only one with a small amount of arachidic acid (0.81%).

It was observed that the Betong and Kanowit samples had no fatty acid compositions compared to the other locations.

In addition, the values obtained in this study indicated that saturated fatty acids (SFAs) and polyunsaturated fatty acids (PUFAs) are the major groups of skin oil as a result of palmitic acid and linoleic acid, while the monounsaturated fatty acids (MUFAs) were in minute quantities. The SFAs ranged from 48.67% to 54.05%, PUFAs ranged from 7% to 47%, and MUFAs ranged from 0% to 43.20% in all of the five localities (Table 2). In this study, no fatty acid content was detected in the dabai skin samples from Betong and Kanowit. The non-detectability of the fatty acids at these two locations does not necessarily mean that they were not present in the samples, but the values may have been below the detection limit of the instrument. More importantly, other overriding factors for the availability of fatty acids in the samples are post-harvesting factors, such as cultural practices; soil type; and climatic conditions, which could play a crucial role in the quantity and quality of fatty acids in dabai fruit. These factors play an important role in the availability of bioactive phytochemicals, nutritional qualities, and antioxidant properties of seed oil. Another possible factor could be differences in genetic characteristics of the seed-bearing plants, which could be the case for the observation in this study.

The production of these fatty acid contents was significantly influenced by the location and geographical factors, which is consistent with the findings of Parcerisa et al. [21]. Studies have shown that geographical factors such as ecology, location, growing conditions, species, altitude, climate, season, soil type, maturity, and harvest period have a significant impact on the fatty acid composition [22]. Other studies like Liu et al. [23], Górnaś et al. [24], He et al. [25], and Sicari et al. [26] have reported a similar trend. Non-detectability of fatty acids could as well be due to the presence of interferences, which could have a high impact in suppressing the fatty acid peaks.

It is also important to observe that the dabai from the Serian geographical location had a higher SFAs (54.05%) content. On the other hand, the dabai skin from Kapit had the highest MUFAs (43.20%) and the lowest PUFAs (7%) content.

Chua et al. [5] reported that the total PUFAs, MUFAs, and SFAs contents of six common dabai skin genotypes from Sibu and Kuching markets, Sarawak, Malaysia, ranged from 13.12% to 41.75%, 43.10% to 48.01%, and 10.24% to 42.62%, respectively. These findings were in agreement with our PUFAs and SFAs values, except for MUFA values.

Similarly, the result showed that the SFA and PUFA are the major groups of dabai pulp fatty acid due to linoleic acid, palmitic acid, and stearic acid, while MUFAs were in minute quantities. The SFAs ranged from 0% to 49.80%, PUFAs ranged from 0% to 43.20%, and MUFAs ranged from 0% to 13% in all of the five localities (Table 3). No fatty acid content was detected in the dabai pulp in Betong and Kanowit

Samples from Serian had higher SFAs (49.80%) and PUFAs (43.20%) contents, while the dabai pulps from Kapit had the lowest SFAs (41.87%) and PUFAs (35.65%) contents.

Azlan et al. [27] reported that the total PUFAs, MUFAs, and SFAs contents of dabai pulp from the Sarawak Agriculture Department using Soxhlet extraction were 12.76%, 42.82%, and 44.43%, respectively. Jelani et al. [28] also reported that the total MUFAs, PUFAs, and SFAs contents of two-mixture clones from the Semongok Agriculture Research Centre were 25.52%, 15.94%, and 59.54%, respectively. The results obtained are in agreement with Azlan et al. [27] and Jelani et al. [28] for SFAs values but not for PUFA and MUFA values. Research on other plants has indicated that the oil of the fruits grown in low-temperature localities contains more unsaturated fatty acids than the fat of fruits grown in hot localities [22]. 

Except for PUFA values, our findings are in line with Shakirin et al. [7], who observed that the total MUFAs of the dabai pulp oil were lower compared to high-fat fruits, such as avocado (65% to 68%) and olive (56% to 86%). The fatty acid profile of dabai pulp contains an oil that is comparable to palm oil [5].

The results showed that the SFA and MUFA are the major groups of dabai kernel fatty acid, containing oleic acid, palmitic acid, and stearic acid, while PUFAs were in minute quantities, except for the dabai kernel samples from the Kapit region. The SFAs ranged from 41.87% to 53.31%, PUFAs ranged from 3.04% to 35.59%, and MUFAs ranged from 21.22% to 47.21% for the three localities (Sibu, Serian, and Kapit), as shown in Table 4. The fatty acid compositions contained in samples have multiple functions, and their bioactivity and mechanism of action highly depends on their composition and environmental factors.

Interestingly, Sibu samples had the highest SFAs (53.31%) and PUFAs (45.91%), while Kapit kernels had the lowest SFAs (41.87%) and PUFAs (35.65%) contents. 

An earlier study by Shakirin et al. [7] reported that the dabai kernel has slightly higher SFAs than MUFAs and PUFAs using Soxhlet extraction. Azlan et al. [27] obtained similar results from their study of the determination of fatty acid content of dabai kernel oil extracted by employing Soxhlet extraction for SFAs and MUFAs values, except for the PUFAs values. They reported that the total of SFAs, MUFAs, and PUFAs content were 60.84%, 35.11%, and 3.78%, respectively. This disagreement for the PUFA values may be due to geographical conditions of the environment, climate, and varieties [22,28,29].

In this study, oleic and palmitic acids accounted for more than 80% of the total fatty acids from Sibu and Serian. In comparison to Ibrahim et al. [30], the oleic and palmitic acids in the tested kernel oil were higher. The kernel oil had a similar fatty acid profile to *C. ovatum*, with oleic and palmitic acids accounting for 32.6–38.2% and 44.4–59.6%, respectively [31]. The SFAs values in the two studied geographical locations (Sibu and Serian) were higher in comparison to *C. album* L., although kernel oil can be characterized as a saturated fatty acid. However, it was less saturated compared to coconut oil [7]. In addition, oil from the dabai kernel contained some saturated fatty acids (lauric and myristic acid), similar to coconut oil.

Multivariate Statistical Analysis of the total amount of fatty acids (%) found in the kernel, pulp, and skin fractions of dabai fruit oil from different localities.

The multivariate statistical analysis revealed interconnected correlation patterns among the fatty acid composition (%) found in the kernel, pulp, and skin fractions of dabai fruit oil from different localities. (Figure 2). 

The spatial arrangement of the fatty acid composition found in the kernel, pulp, and skin fractions of dabai fruit oil from Serian and Kapit in the double-positive quadrant signified the presence of interrelated fatty acids (Figure 2a). Moreover, clustering of the fatty acid composition exhibited a mixed pattern in describing the variation of the three localities. Furthermore, the presence of a fairly distinct fatty acid composition in Sibu, Serian, and Kapit was highlighted by their isolated spatial location as well as the correlation matrix (Table 5). It was interesting that the clustering analysis using a hierarchical method (Figure 2b) showed a concordant pattern with the PCA analysis. The fatty acid composition of dabai in the three localities was grouped into various clusters, according to their proximities. The proximity heat map (Figure 2c) corroborated the correlation matrix, demonstrating a similar fatty acid composition in the three localities of dabai.

### 3.2. Antioxidant Activity Measured Using DPPH

The values of DPPH free radical scavenging activity at 1000 µg/mL for skin, pulp, and kernel fractions of dabai fruit oils from the different geographical locations are shown in Table 6. At a concentration of 1000 µg/mL, the skin fraction of dabai from the Sibu locality exhibited the highest percentage (69.84 ± 0.14%) compared to the other fruit lipids.

The results of the test of variance of the antioxidant bioactivity in terms of the EC_50_ of the skin, pulp, and kernel fractions of dabai extracts showed that the effect of geographical location was significant (*p* < 0.05) (Table 6).

These data agreed with Rashid et al. [32], who reported that dabai skin is the principal antioxidant source because of it being rich in phenolic contents. In addition, the EC_50_ value for a skin fraction of dabai from Sibu was 198.76 ± 1.06 µg/mL; thus, the strength for this extract to act as a DPPH scavenger was higher compared to the skin fractions of dabai from Serian, Betong, Kapit, and Kanowit (Table 7). However, the EC_50_ values for other dabai lipid extracts could not be obtained due to their very low percentage of antioxidant activity (Table 8). These findings were in line with the works of Jelani et al. [28] and Rashid et al. [32]. It may be due to the low total phenolic, flavonoid, and anthocyanin contents.

The result confirmed that the EC_50_ values varied from 320.64 ± 1.09 to 198.76 ± 1.06 µg/mL. The EC_50_ value of the standard ascorbic acid solution was lower at 18.59 ± 1.06 µg/mL compared to EC_50_ values for all dabai lipid extracts, indicating that ascorbic acid had the strongest power to scavenge DPPH radicals. Since EC_50_ is inversely associated with the anti-radical ability of the compounds, the lower the EC_50_, the higher the antioxidant activity [33]. Based on the outcomes of this study, the effect of the geographical location on the antioxidant ability of the skin fraction was very profound, which is consistent with the findings of Dastoor et al. [34], Koohsari et al. [35], and Zargoosh et al. [36]. The skin fraction of the dabai from Sibu, with the lowest EC_50,_ had the highest antioxidant bioactivity. The natural antioxidant components contained in the fruit extract fractions have multiple functions, and their bioactivity and mechanism of action greatly depend on their geographical and cultivation conditions, since these conditions affect the synthesis of the formation of secondary bioactive compounds in the plant. Plant geographical location, as a result of environmental differences, weather differences, and soil conditions, can influence the formation of secondary bioactive compounds in the plant. In addition, it might also be due to salinity induced by the metabolic systems [36]. The relative geographical habitat of this locality is directly associated with the increase in antioxidant activity [37]. 

The topography could have also contributed to the antioxidant property of the collected samples from Serian and Sibu. This may be due to reduced temperature and increased exposure to ultraviolet radiation of the organ responsible for the synthesis of antioxidant elements.

Multivariate Statistical Analysis of the values of DPPH free radical scavenging activity at 1000 µg/mL for skin, pulp, and kernel fractions of dabai fruit oils from the different geographical locations.

The multivariate statistical analysis revealed interconnected correlation patterns among the values of DPPH free radical scavenging activity at 1000 µg/mL for skin, pulp, and kernel fractions of dabai fruit oils from the different geographical locations (Figure 3). 

The spatial arrangement of the DPPH (%) found in the kernel, pulp, and skin fractions of dabai fruit oil from Sibu, Serian, Kapit, Betong, and Kanowit in the double-positive quadrant signified the presence of interrelated DPPH (%) (Figure 3a). Moreover, clustering of DPPH (%) exhibited a mixed pattern in describing the variation of the five localities. Furthermore, the presence of fairly distinct DPPH (%) in Sibu, Serian, Kapit, Betong, and Kanowit was highlighted by their isolated spatial location as well as the correlation matrix (Table 7). It was interesting that the clustering analysis using a hierarchical method (Figure 3b) showed a concordant pattern with the PCA analysis. The values of DPPH free radical scavenging activity at 1000 µg/mL for the skin, pulp, and kernel fractions of dabai fruit oils from the five localities were grouped into various clusters according to their proximities. The proximity heat map (Figure 3c) corroborated the correlation matrix, demonstrating similar DPPH (%) in the five localities of dabai.

### 3.3. Brine Shrimp Lethality Bioassay (BSLA)

The results of the brine shrimp lethality (BSL) test at various concentrations for *Canarium* spp. oil extracts, as well as their plots of percentage mortality versus concentrations, are shown in Table 9, respectively. An LC_50_ value of less than 1000 µg/mL is toxic, whereas an LC_50_ value of higher than 1000 µg/mL is non-toxic [38,39].

The *Artemia* species are very useful and suitable for toxicity evaluation of bioactive substances in crude lipid fruit fractions. The BSL procedure is one of the most useful, reliable, and routine assays in the laboratory. This assay has been used for toxicity screening of some pesticides, heavy metals, food additives, and pharmaceutical compounds [16,17,18,19]. Due to its low cost, simplicity, and high sensitivity, the BSL assay has been receiving great attention from many researchers [16,17,18,19] 

Natural deaths (mortalities), which were evaluated in blank seawater and wells treated with the positive control only, usually did not exceed 25%. This seemed to be a result of a lack of oxygen because most of the *A. salina* nauplii did not survive beyond 48 h of the assay [16,17,18,19]. In this regard, factors like age, composition, pH, the salinity of the matrix, and the temperature of larvae are effective factors in natural mortality [16].

During the study, no feed or air was needed because feeding brine shrimps with dry yeast suspension during the toxicity assessment is considered insignificant [16,17,18,19].

The results showed that the skin fraction had higher lethality concentrations (LC_50_) than other fractions at all geographical locations. There was a significant difference (*p*< 0.05) in the effect of the geographical location on the toxicity of the three fractions against *A. salina* (Table 9). Basri et al. [11] suggest that the contents of saponins, terpenoids, tannins, and flavonoids influence toxicity. This is in line with Khoo et al. [40,41], who reported that the polyphenol components found in the dabai skin contain saponin, flavonoid, carotenoids, and anthocyanins; the fruit pulp contains anthocyanin and carotenoids; and the kernel contain α tocopherols, γ- tocopherols, and flavonoid [42]. Therefore, it was suspected that the cytotoxicity level against *A. salina* in the skin fraction extracts was induced by saponin, flavonoid, and anthocyanins contents, since the concentration of these three molecules might be richer in the skin compared to those in the kernel and pulp. Anthocyanin is known to possess antimicrobial, anti-obesity, anti-inflammatory, antidiabetic, and anticancer effects, as well as the prevention of chronic diseases [43]. The anticancer and other mechanisms of anthocyanins are based on their antioxidant potential, which is linked to their ability to scavenge free radicals, inhibit the enzymes involved in ROS formation, and prevent the oxidation of extracellular and cellular biomolecules [44].

Flavonoids are found in all growing parts of the plant, being reported as the most abundant plant pigment, along with carotenoids and chlorophyll, also providing taste and fragrance to seeds, fruits, and flowers, which make them attractive to other organisms [45,46]. These bioactive compounds are also among the largest groups of secondary metabolites [47]. Besides their significance in plants, flavonoids are crucial for human health due to their significant pharmacological bioactivities. However, the production of these compounds in the pulp and kernel of dabai fraction extracts, although under the control of genetic factors, is remarkably affected by the geographical location. Investigations have revealed that environmental factors. such as geographical location. can change the level of phytochemical components [35,48]. Climatic factors, like lower temperatures, high relative humidity. as well as high rainfall. might be a reason for the higher compound contents in different geographical locations [49,50,51].

Higher LC_50_ values (>1000 mg/L) were observed in the kernel, pulp. and skin fraction extracts from the different localities. This indicates that the oil extract might be safe for pharmaceutical uses. More evidence (e.g., clinical trials) might be needed to substantiate its safety status. Hence, dabai cultivation in the Sarawak region, specifically in Sibu and Serian, could serve as a major contributor to the economy. Dabai has good fatty acid profiles, remarkable antioxidant activity, and is non-toxic to human life.

## 4. Conclusions

The extracts of the edible fractions from dabai fruits (kernel, oil, and pulp) collected at five different geographical locations in Sarawak showed different fatty acid compositions, antioxidants, and cytotoxicity. Among the locations, the Sibu and Serian samples’ extracts showed a higher fatty acid composition and antioxidant potential. They also exhibited higher mortality of the brine shrimp than the samples from other locations (Betong, Kapit, and Kanowit). A significant correlation between antioxidant properties and cytotoxicity showed that certain phenolic bioactive compounds were the main contributors to the cytotoxicity properties of dabai fruits (kernel, oil, and pulp). Regarding the differences between the fraction extracts, we concluded that the geographical locations of samples influenced the fatty acid composition and phytochemicals of the plant. Furthermore, this study contributes to the origin effect, which has not been thoroughly studied, despite the reported effects of genotypes. Climate change might have also affected the reported values in this study compared to previous reports.

For any future application of the extract of the edible fraction of *C. odontophyllum* L. fruits as an antifungal biopesticide in organic agriculture, the location of the sample’s collection needs to be included in the standardization of the formulations. The management of the geographical differences is essential to any successful agricultural and bioengineering industrial application.

## Figures and Tables

**Figure 1 molecules-27-03840-f001:**
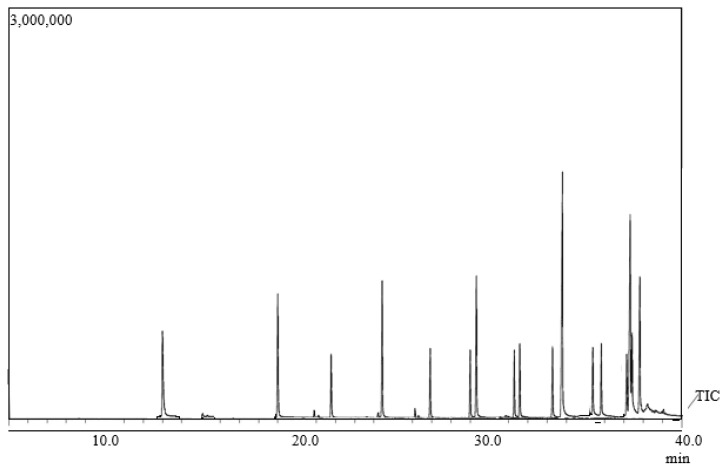
GC-MS chromatogram for the mixture of FAME standards.

**Figure 2 molecules-27-03840-f002:**
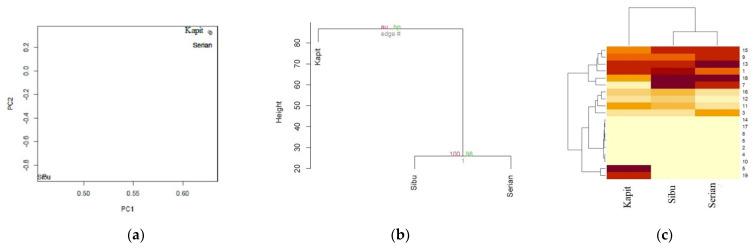
The multivariate statistical analysis revealed interconnected correlation patterns among the fatty acid composition (%) found in the kernel, pulp, and skin fractions of dabai fruit oil from different localities. (**a**) PCA based on the correlation matrix of fatty acids (%); (**b**) A consensus tree of the relationships among Sibu, Serian, and Kapit in the fatty acids (%). The Pvclust package in R was used to cluster these traits according to the Euclidean distance matrix. The numbers at the forks were the percentages of approximately unbiased (AU; in red) p-values and bootstrap probabilities (BP, in green), estimated from 1000 bootstrapping samples; (**c**) A heatmap showing cluster groups in both the fatty acids (%) and different localities (Sibu, Serian, and Kapit). The numbers on the right side represent the codes for 19 fatty acids (%) found in the kernel, pulp, and skin fractions within three localities that showed a mixed pattern.

**Figure 3 molecules-27-03840-f003:**
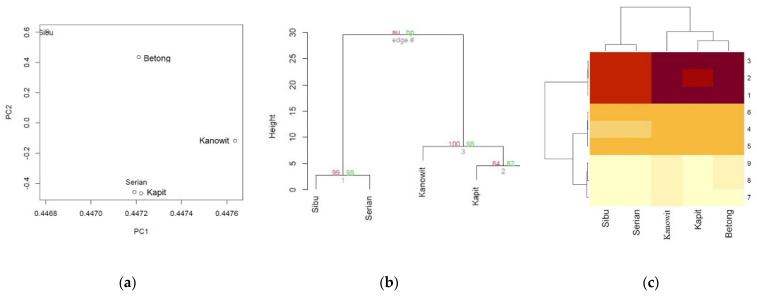
Multivariate statistical analysis revealed interconnected correlation patterns among the values of DPPH free radical scavenging activity at 1000 µg/mL for skin, pulp, and kernel fractions of dabai fruit oils from the different geographical locations. (**a**) PCA based on the correlation matrix of DPPH (%); (**b**) A consensus tree of the relationships among Sibu, Serian, Kapit, Betong, and Kanowit in the DPPH (%). The Pvclust package in R was used to cluster these traits according to the Euclidean distance matrix. The numbers at the forks were the percentages of approximately unbiased (AU; in red) p-values and bootstrap probabilities (BP, in green) estimated from 1000 bootstrapping samples; (**c**) A heatmap showing cluster groups in both the DPPH (%) and different localities (Sibu, Serian, Kapit, Betong, and Kanowit). The numbers on the right side represent the codes for 9 DPPH free radical scavenging activity at 1000 µg/mL for skin, pulp, and kernel fractions of dabai fruit oils from the different geographical locations, which showed a mixed pattern.

**Table 1 molecules-27-03840-t001:** Retention times for identified components of FAME standards analyzed by GC-MS.

No.	Fatty Acid (as Methyl Esther)	IUPAC	Retention Times, t_Rx_ (min)
1	Undecanoic acid methyl ester	C11:0	13.14
2	Lactic acid methyl ester	C12:0	18.12
3	Tridecanoic acid methyl ester	C13:0	20.62
4	Myristic acid methyl ester	C14:0	23.65
5	Pentadecanoic acid methyl ester	C15:0	26.71
6	Palmitoleic acid methyl ester	C16:1	29.01
7	Palmitic acid methyl ester	C16:0	29.43
8	Heptanoic acid methyl ester	C17:0	31.76
9	Linolenic acid methyl ester	C18:3n3	33.17
10	Linoleic acid methyl ester	C18:2n6c	33.64
11	Oleic acid methyl ester	C18:1n9c	35.69
12	Stearic acid methyl ester	C18:0	36.26
13	Arachidonic acid methyl ester	C20:4n6	37.21
14	Eicosadienoic acid methyl ester	C20:2	37.86

**Table 2 molecules-27-03840-t002:** Fatty acid profiles (%) found in the skin fractions of *Canarium odontophyllum* L. fruit from different localities.

	Sibu	Serian	Kapit	Betong	Kanowit
SFA					
C16:0	39.66 ± 0.14 ^bc^	27.90 ± 0.67 ^a^	36.49 ± 0.33 ^b^	ND	ND
C17:0	ND	ND	1.81 ± 0.21 ^a^	ND	ND
C18:0	9.01 ± 0.32 ^a^	23.42 ± 0.98 ^c^	11.50 ± 0.11 ^b^	ND	ND
C20:0	ND	2.73 ± 0.83 ^a^	ND	ND	ND
Total:	48.67 ± 0.21 ^a^	54.05 ± 0.77 ^a^	49.80 ± 0.59 ^a^	ND	ND
MUFA					
C18:1n9c	ND	ND	43.20 ± 0.32 ^a^	ND	ND
C18:1n9t	ND	ND	ND	ND	ND
Total:	ND	ND	43.20 ± 0.32 ^a^	ND	ND
PUFA					
C18:2n6c	47.04 ± 0.09 ^c^	35.65 ± 0.25 ^b^	7.00 ± 0.50 ^a^	ND	ND
C20:2	ND	0.81 ± 0.11 ^a^	ND	ND	ND
Total:	47.04 ± 0.09 ^c^	36.46 ± 0.35 ^b^	7.00 ± 0.50 ^a^	ND	ND

The triplicate measurements were performed with the same 100 g of each fraction sample. Values are presented as mean ± SD (*n* = 3). Non-identical superscripts in the same row represent significant differences at *p* < 0.05. SFA = saturated fatty acid; PUFA = polyunsaturated fatty acids; MUFA = mono-unsaturated fatty acids; ND = not detected.

**Table 3 molecules-27-03840-t003:** Total amount of fatty acids (%) found in the pulp fractions of dabai fruit oil from different localities.

	Sibu	Serian	Kapit	Betong	Kanowit
SFA					
C16:0	29.86 ± 0.99 ^ab^	36.49 ± 0.76 ^b^	27.90 ± 0.88 ^a^	ND	ND
C17:0	ND	1.81 ± 0.77 ^a^	ND	ND	ND
C18:0	16.77 ± 0.11 ^ab^	11.50 ± 0.25 ^a^	23.42 ± 0.55 ^b^	ND	ND
Total:	46.63 ± 0.15 ^ab^	49.80 ± 0.33 ^b^	41.87 ± 0.45 ^a^	ND	ND
MUFA					
C18:1n9c	13.38 ± 0.97 ^bc^	7.00 ± 0.86 ^a^	10.30 ± 0.75 ^b^	ND	ND
Total:	13.38 ± 0.15 ^bc^	7.00 ± 0.35 ^a^	10.30 ± 0.55 ^b^	ND	ND
PUFA					
C18:2n6c	38.58 ± 0.66 ^ab^	43.20 ± 0.73 ^b^	35.65 ± 0.79 ^a^	ND	ND
C20:2	1.40 ± 0.12 ^a^	ND	ND	ND	ND
Total:	39.98 ± 0.06 ^ab^	43.20 ± 0.58 ^b^	35.65 ± 0.13 ^a^	ND	ND

The triplicate measurements were performed with the same 100 g of each fraction sample. Results are expressed as mean value ± SD (*n* = 3). Non-identical letters in the same row indicate significantly different values at *p* < 0.05. SFA = saturated fatty acid; PUFA = polyunsaturated fatty acids; MUFA = mono-unsaturated fatty acids; ND = not detected.

**Table 4 molecules-27-03840-t004:** Total amount of fatty acids (%) found in the kernel fractions of dabai fruit oil from different localities.

	Sibu	Serian	Kapit	Betong	Kanowit
SFA					
C16:0	35.82 ± 0.33 ^b^	38.32 ± 0.44 ^bc^	26.24 ± 0.55 ^a^	ND	ND
C18:0	16.68 ± 0.48 ^b^	10.64 ± 0.37 ^a^	15.63 ± 0.26 ^b^	ND	ND
C20:0	0.81 ± 0.59 ^a^	ND	ND	ND	ND
Total:	53.31 ± 0.80 ^c^	48.96 ± 0.91 ^b^	41.87 ± 0.39 ^a^	ND	ND
MUFA					
C18:1n9c	45.91 ± 0.28 ^bc^	43.65 ± 0.19 ^b^	21.22 ± 0.37 ^a^	ND	ND
Total:	45.91 ± 0.28 ^bc^	43.65 ± 0.19 ^b^	21.22 ± 0.37 ^a^	ND	ND
PUFA					
C18:2n6c	3.04 ± 0.13 ^a^	3.83 ± 0.26 ^a^	35.59 ± 0.39 ^b^	ND	ND
Total:	3.04 ± 0.13 ^a^	3.83 ± 0.26 ^a^	35.59 ± 0.39 ^b^	ND	ND

The triplicate measurements were performed with the same 100 g of each fraction sample. Results are expressed as mean ± SD (*n* = 3). Non-identical letters in the same row indicate significantly different values at *p* < 0.05. SFA = saturated fatty acid; PUFA = polyunsaturated fatty acids; MUFA = mono-unsaturated fatty acids; ND = not detected.

**Table 5 molecules-27-03840-t005:** Correlation Coefficient Matrix of the fatty acids (%) found in the kernel, pulp, and skin. Fractions of dabai fruit oil from different localities.

	Sibu	Serian	Kapit
Sibu	1.0000		
Serian	0.9387524	1.0000	
Kapit	0.4684037	0.4738977	1.0000

Note: *p*-values for statistical tests are in parentheses.

**Table 6 molecules-27-03840-t006:** DPPH free RSA at 1000 µg/mL for three different fractions of extracted dabai fruit lipids of different localities with standard ascorbic acid.

Fruit Lipids	Percentages of DPPH Scavenging Activity at 1000 µg/mL (±S.D.%)
*C. odontophyllum* L. skin fraction	
Sibu	69.84 ± 0.01
Serian	65.08 ± 0.30
Kapit	57.38 ± 0.51
Betong	60.09 ± 0.12
Kanowit	53.61 ± 0.14
*C. odontophyllum* L. pulp fraction	
Sibu	29.68 ± 0.52
Serian	29.49 ± 0.17
Kapit	30.08 ± 0.58
Betong	30.08 ± 0.38
Kanowit	31.41 ± 0.57
*C. odontophyllum* L. kernel fraction	
Sibu	12.00 ± 0.18
Serian	11.37 ± 1.76
Kapit	13.36 ± 0.59
Betong	14.97 ± 0.59
Kanowit	15.60 ± 0.14
Ascorbic acid (Standard reference)	96.92 ± 0.00

**Table 7 molecules-27-03840-t007:** Correlation Coefficient Matrix of the values of DPPH free radical scavenging activity at 1000 µg/mL for the skin, pulp, and kernel fractions of dabai fruit oils from the different geographical locations.

	Sibu	Serian	Kapit	Betong	Kanowit
Sibu	1.0000				
Serian	0.9991089	1.0000			
Kapit	0.9981758	0.9967917	1.0000		
Betong	0.9953717	0.9940903	0.9988797	1.0000	
Kanowit	0.9956498	0.9938128	0.9992088	0.9997455	1.0000

Note: *p*-values for statistical tests are in parentheses.

**Table 8 molecules-27-03840-t008:** EC_50_ values of different fractions of *C. odontophyllum* L. lipids from different localities.

Lipid Sample	EC_50_ Value (±S.D. µg/mL)
Skin fraction	
Sibu	198.76 ± 1.06 ^c^
Serian	214.88 ± 1.22 ^bc^
Kapit	298.24 ± 1.08 ^ab^
Betong	289.08 ± 1.07 ^b^
Kanowit	320.64 ± 1.09 ^a^
Pulp fraction	
Sibu	N.D.
Serian	N.D.
Kapit	N.D.
Betong	N.D.
Kanowit	N.D.
Kernel fraction	
Sibu	N.D.
Serian	N.D.
Kapit	N.D.
Betong	N.D.
Kanowit	N.D.
Ascorbic acid (control)	18.59 ± 1.06 ^d^

The triplicate measurements were performed with the same 100 g of each fraction sample. Values are presented as mean ± SD (*n* = 3). Non-identical superscripts in the same row represent significant differences at *p* < 0.05. ND: not detected.

**Table 9 molecules-27-03840-t009:** Average death of *A. salina* nauplii at different concentrations of dabai extracts of different localities.

Sample Extracts	Average Death of *A. salina* Nauplii						LC_50_ (µg/mL)
	Concentration (ppm)						
	1	10	50	100	250	500	
Skin							
Sibu	0.00 ± 0.00	0.00 ± 0.00	1.00 ± 1.00	0.67 ± 1.15	1.00 ± 0.73	1.67 ± 2.08	1387.22
Serian	0.00 ± 0.00	1.00 ± 1.73	1.33 ± 1.53	1.33 ± 1.15	1.67 ± 0.58	1.67 ± 1.15	1441.56
Kapit	0.00 ± 0.00	1.00 ± 1.00	1.33 ± 2.31	1.00 ± 1.73	1.00 ± 0.00	2.00 ± 1.00	1754.74
Betong	0.00 ± 0.00	1.67 ± 2.08	0.33 ± 0.58	1.00 ± 1.00	1.33 ± 0.58	1.67 ± 1.53	1528.52
Kanowit	0.00 ± 0.00	0.00 ± 0.00	1.00 ± 1.00	0.67 ± 1.15	0.33 ± 0.58	1.33 ± 0.58	1842.65
Pulp							
Sibu	0.67 ± 1.15	1.00 ± 1.00	1.33 ± 1.15	0.67 ± 0.58	1.33 ± 1.53	2.00 ± 1.00	7182.10
Serian	0.00 ± 0.00	0.00 ± 0.00	1.00 ± 1.00	0.67 ± 1.15	1.33 ± 1.53	2.00 ± 1.00	6249.25
Kapit	0.00 ± 0.00	0.00 ± 0.00	0.33 ± 0.58	1.00 ± 1.00	1.33 ± 1.53	2.00 ± 1.00	5061.03
Betong	0.67 ± 1.15	0.67 ± 0.58	1.00 ± 0.00	2.00 ± 1.00	1.67 ± 1.53	2.67 ± 2.08	7196.53
Kanowit	0.33 ± 0.58	0.67 ± 1.15	1.00 ± 1.00	0.67 ± 0.58	2.00 ± 0.00	2.67 ± 1.53	4192.32
Kernel							
Sibu	0.33 ± 0.58	1.00 ± 0.00	1.33 ± 0.58	2.33 ± 0.58	3.00 ± 0.00	3.67 ± 0.58	2742.65
Serian	1.00 ± 1.73	0.67 ± 0.58	2.00 ± 1.00	3.00 ± 0.00	2.67 ± 1.53	4.00 ± 1.00	2578.46
Kapit	0.67 ± 1.15	1.33 ± 1.53	2.00 ± 1.73	2.33 ± 0.58	3.00 ± 2.65	4.33 ± 1.15	2928.31
Betong	1.00 ± 1.00	1.33 ± 0.58	2.00 ± 1.00	3.33 ± 2.08	3.00 ± 1.00	4.67 ± 2.08	2261.04
Kanowit	0.00 ± 0.00	0.33 ± 0.58	1.33 ± 0.58	1.67 ± 0.58	2.67 ± 0.58	3.00 ± 0.00	1954.74
Thymol (positive control)	5.00 ± 1.00	7.33 ± 0.58	10.00 ± 0.00	10.00 ± 0.00	10.00 ± 0.00	10.00 ± 0.00	10.32
DMSO + seawater (negative control)	0	0	0	0	0	0	-

## Data Availability

Not available.

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
