# Peer review of "Fatty Acid Profile and Antioxidant Capacity of Dabai (Canarium odontophyllum L.): Effect of Origin and Fruit Component"

_molecules, 2022, doi:10.3390/molecules27123840_

Round 1

Reviewer 1 Report

2.Materials and methods

Line 174 please fix a typo “probity???”. I believe the authors mean statistical tests

Please report LOD (limit of detection) for each essay

3.Results

For all tables, please clarify in the footnote that the triplicates were performed with the same 100 g sample.

For Tables 2-4, please use BLD (below limit of detection) or ND (not detected) in place of NA when the fatty acids were not detected in the samples. For Betong and Kanowit, were the samples not available or were the fatty acids not detected? If the latter were true, it would be very questionable to see a very wide range of values within the same species of plants. If there were no fats at all, then what kind of organic matter does the fruit composed of? It should not be purely water and carbohydrate since it is generally known that the dabai fruit has a creamy texture. Did the texture of the fruits from these two regions significantly differ from the other three regions?

The values were also in the similar range among Sibu, Serian, and Kapit, so it is very questionable that fatty acids could not be detected in the other two regions.

For Tables 2-4, please clarify that the % reported were % of dry weight, not % of total fatty acids

Line 282 I’m unclear what the definition of “toxic fatty acids” is in this context. Perhaps the authors mean trans-fatty acids?

Table 5 please indicate that the values were averaged from triplicate experiment samples

Line 292, the value in the text says 68.84% but the table says 69.84%. Please also move Lines 291-292 to before Table 5.

Line 391, please revise the statement. The evidence with the BSLA is not sufficient to state that it is “”non-toxic to human life”, since more evidence (e.g. clinical trials) is needed.

Could climate changes affect the reported values in this manuscript compared to previous reports?

4.Conclusion

Line 396 it is unclear what a “superior” fatty acid composition is. Please revise this sentence.

Author Response

Please see rebuttal letter for reviewer 1 comment attached below

Reviewer 2 Report

The authors aimed to evaluate the effect of origin (four locations) and fruit component (skin, pulp, kernel) of Dabai (Canarium odontophyllum L) fruit on its fatty acid profile (by GC), antioxidant capacity (DPPH) and toxicity (brine shrimp larvae). The experimental design was adequate, and results indicate a greater influence of fruit part rather than origin on the assayed parameters. Although the data is quite relevant and unique, the scientific value of the study could be improved if performing some adjustments to the manuscript. 

General

  • The readability and syntax of the manuscript will be substantially improved if it is reviewed by a formal translation agency or by a native English spoken person.
  • Certain section (e.g. references) should be formatted according to Molecules´ guidelines.

Title. Quite long. Suggestion: Fatty acid profile and antioxidant capacity of Dabai (Canarium odontophyllum L.): Effect of origin and fruit component

Abstract. It should be more concise and quantitative without sacrificing important differential results.

Introduction & conclusion.  A quick search of information in academic search engines confirms that the origin effect has not been thoroughly studied, even though the effects of genotypes (possibly associated with their origin) and part of the fruit are in the reported parameters (http://jtafs.mardi. gov.my/jtafs/43-1/Dabai%20fruit.pdf, https://journals.hh-publisher.com/index.php/AAFRJ/article/view/272). The authors are suggested to highlight this singularity even more. Do the same with the conclusion section.

Results & discussion.

  • The authors should start this section with a brief introduction to the technological and nutritional importance of Dabai fruit and extracted oil and the lack of information on the effect of production origin.
  • Lines 177-179 could be included as footnotes in Figure 1 and Table 1 (as supplementary materials, Figure S1 and Table S2).
  • The statement between lines 191-192 is not clear or is incorrect (these fruits do not have fatty acids??)
  • The statement between lines 358-361 is incoherent, what does this mean?.
  • By having many parameters evaluated for the same experimental treatments (region x fruit part), the authors have a unique opportunity to use inductive clustering methods (e.g. HCA, PCA, PLS-DA) to study the individual and additive effects of factors under study and to propose treatments with greater and lesser phytochemical richness. It is recommended to carry them out and to include useful figures in both bodytext and supplementary material.

Figures (F) & Tables (T).

  • Please improve the resolution of all figures (≥300 dpi or more), format all tables according to Molecules guidelines, and improve any footnotes in both (F&T).
  • The way you used superscript letters to denote statistical differences between treatments for the same parameter is incorrect, use the conventional format (e.g. Table 2, for C16:0 = Sibua - kapitb - Serianc).
  • The term “NA indicates data not available” (Tables 2-4) raises doubts as to the sensitivity/specificity of the method because the natural variability, although affected by many factors including analytic factors, normally ranges between 10%-50% but not to the point of being undetected (e.g. only the Kapit variety has MUFAs?).
  • F1 and T1 could be submitted as supplementary material (Figure/Table S1), indicating in the figure the corresponding peak number (as referred to in the table) and in the table change “composition” as “IUPAC” and “Compound” as “Fatty acid (as methyl Esther)”.
  • Authors could eliminate figures 2-3 since the same information is reported in Table 7.

References. Please format according to journal guidelines and decrease the number of old citations (≥ 10y) to say 25% or less

Author Response

Please see attached rebuttal letter for reviewer 2 comments

Round 2

Reviewer 1 Report

The authors have satisfied my previous comments and suggestions. Two minor suggestions remain.

Methods and Materials. Please briefly discuss why the limit of detection cannot be determined in this GC-MS method.

Line 350 the text says "69.84±0.14%" but the table says "69.84±0.14%"

Reviewer 2 Report

The authors have modified their manuscript satisfactorily